# Exposing 24-hour cycles in bile acids of male humans

Adesola T. Bello[1,7], Magali H. Sarafian[2,7], Elizabeth A. Wimborne[3], Benita Middleton[4], Victoria L. Revell[4,5], Florence I. Raynaud [6], Namrata R. Chowdhury[4], Daan R. van der Veen [4], Debra J. Skene [4] & Jonathan R. Swann [1,3] ✉

Bile acids are trans-genomic molecules arising from the concerted metabolism of the human host and the intestinal microbiota and are important for digestion, energy homeostasis and metabolic regulation. While diurnal variation has been demonstrated in the enterohepatic circulation and the gut microbiota, existing human data are poorly resolved, and the influence of the host circadian system has not been determined. Using entrained laboratory protocols, we demonstrate robust daily rhythms in the circulating bile acid pool in healthy male participants. We identify temporal relationships between bile acids and plasma lipids and show that these relationships are lost following sleep deprivation. We also highlight that bile acid rhythmicity is predominantly lost when environmental timing cues are held constant. Here we show that the environment is a stronger determinant of these temporal dynamics than the intrinsic circadian system of the host. This has significance for the intimate relationship between circadian timing and metabolism.

The hologenome theory of evolution considers that the overall holobiont, encompassing the genome of the host and its resident microbiota, evolved to the demands of its environment to collectively define its fitness. A key component of this fitness is the ability to anticipate daily changes in the environment rather than respond to them[1]. For example, daily rhythms in hormones and metabolites help to prepare the metabolic readiness of the organism for extrinsic fluctuations. Such changes can be a result of both the endogenous circadian timing system driven by intrinsic clocks and/or exogenous factors (e.g., light/dark; feeding/fasting; sleep/wake). While daily rhythms have largely been studied in the host only, there is increasing evidence that daily variation occurs in the populations and functions of the resident gut microbiota[2–4]. However, little is known about the drivers of these modulations or their cross-talk and impact on the host.

Bile acids are an important class of trans-genomic metabolites arising from the combined metabolism of the mammalian host and the intestinal microbiota. These metabolites have a key role in the digestion of lipids and lipid-soluble vitamins and act as signaling molecules binding to a range of nuclear receptors expressed throughout the host. Through this binding, bile acids can influence lipid, glucose, and drug metabolism, energy homeostasis, and immune responses, as well as their own synthesis and transport[5–8]. As bile acids are secreted into the gut upon feeding, and given their known endocrine functions, it is hypothesized that these molecules may contribute to priming the host metabolic system at the appropriate time to process dietary intake. Indeed, previous human studies have demonstrated diurnal variation in specific bile acids; however, the resolution of such studies has been limited both temporally (e.g., ≤2 sampling points) and compositionally (i.e., small number of bile acids measured), and did not remove

[1]Section of Nutrition, Department of Metabolism, Digestion and Reproduction, Imperial College London, London, UK. [2]Bioaster Microbiology Technology Institute, Lyon, France. [3]School of Human Development and Health, Faculty of Medicine, University of Southampton, Southampton, UK. [4]Chronobiology Section, Faculty of Health and Medical Sciences, University of Surrey, Guildford, Surrey, UK. [5]Surrey Sleep Research Centre, Faculty of Health and Medical Sciences, University of Surrey, Guildford, UK. [6]Centre for Cancer Drug Discovery, Division of Cancer Therapeutics, The Institute of Cancer Research, London, UK. [7]These authors contributed equally: Adesola T. Bello, Magali H. Sarafian. ✉e-mail: j.swann@soton.ac.uk

external factors[9–11]. Interestingly, bile acids also possess antimicrobial properties, providing protection against infections, and can exert selection pressures on the gut microbial community structure[12,13]. Modifications of these co-metabolites by both the host and the microbiota can change their physicochemical, signaling, and antimicrobial properties with implications at the host and microbial level[14]. As such, bile acids may represent an important mechanism in the holobiont for the reciprocal tuning of host and microbial daily fluctuations. Dysregulation of this pan-kingdom communication may, therefore, have ramifications for lipid processing, energy homeostasis, and metabolic resilience.

Laboratory protocols undertaken in entrained conditions and in constant routine conditions can characterize daily rhythms and intrinsically driven circadian rhythms in variables, respectively. Whilst robust diurnal and circadian rhythms in the human plasma metabolome have been described[15–17], the profiling of bile acid signatures in these entrained and circadian protocols has not yet been performed. Moreover, the effect of total sleep deprivation on bile acid rhythmicity has not been studied. This has potential significance given the well-established links between shift work, metabolic dysregulation, and metabolic syndrome, as well as the role of bile acids in these processes[18].

In this study, using both entrained and constant routine protocols, we demonstrate a robust daily rhythm in the overall circulating bile acid pool of healthy male participants, with phase variation in the host-derived conjugated and microbiota-derived unconjugated bile acids. We highlight relationships between daily time courses in bile acids and plasma lipids. We show that sleep deprivation exerts a modest impact on the timings of these rhythms but results in the loss of synchronicity between the circulating bile acids and lipidome. Importantly, the rhythmicity of the bile acids is lost in constant routine conditions when the environmental timing cues are removed or kept constant. These findings establish that the environment exerts a stronger influence on the circulating bile acids, key trans-genomic signaling molecules that impact both the host and microbiota, than the intrinsic circadian timing system of the host.

## Results
### Circulating bile acids follow a daily cycle with conjugated bile acids preceding unconjugated bile acids
To investigate if the bile acid pool followed a daily cycle, a human-entrained laboratory study was performed in which light exposure and behavioral patterns (sleep/wake; feeding/fasting) were strictly controlled (Fig. 1A). Here, 15 healthy male participants (mean ± SD, age 23.7 ± 5.4 years; BMI 24.9 ± 2.6 kg/m²; Supplementary Data 1) remained under controlled laboratory conditions regarding sleep, environmental light, and posture during a 24-hour wake/sleep cycle followed by a 24-hour period of wakefulness (total sleep deprivation). Standardized meals were provided at 07:00, 13:00, and 19:00 h with a snack at 22:00 h with a total of 2500 calories per day divided into

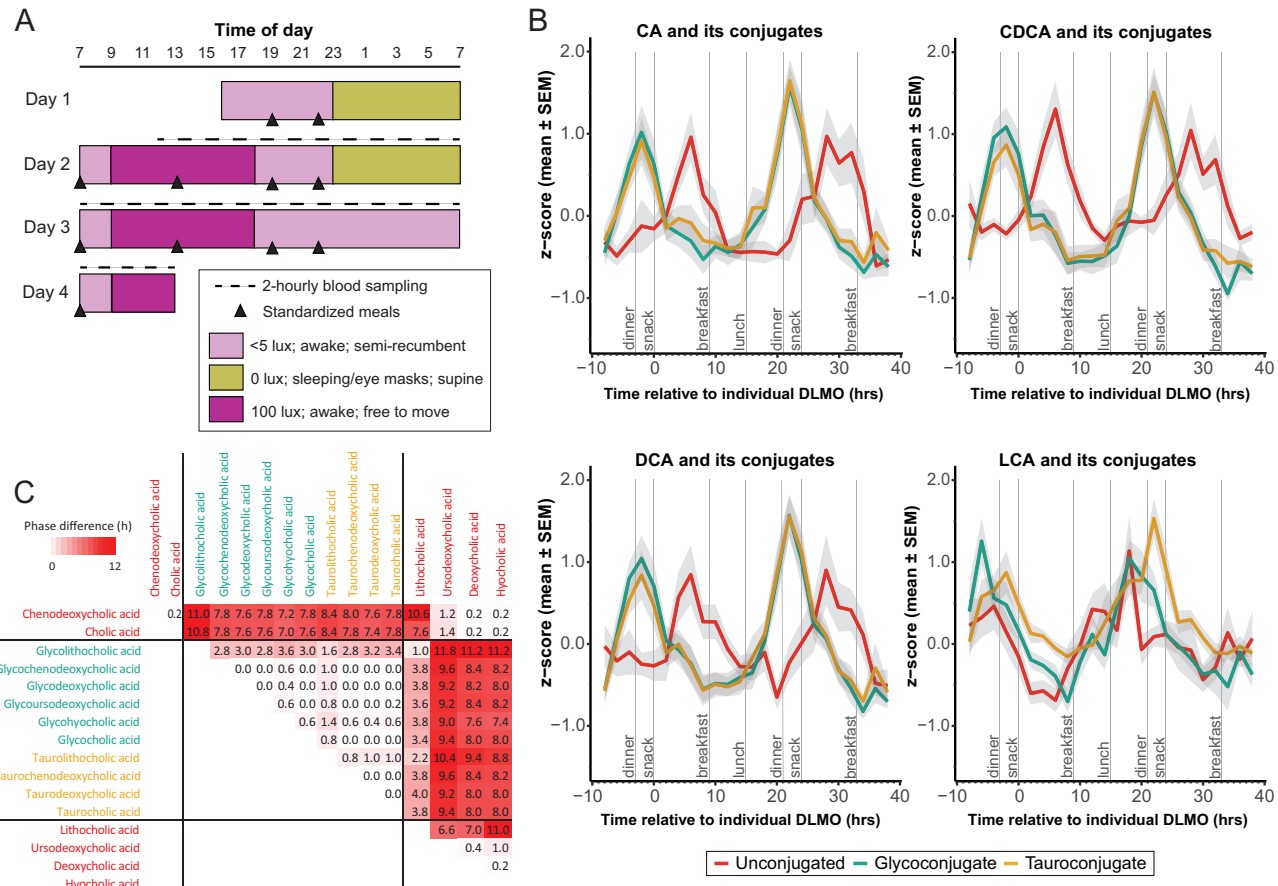

**Fig. 1 | Circulating bile acids follow a daily cycle. A** Study design for the entrained human study (24 h sleep/wake) followed by 24 h of wakefulness. **B** Daily patterns in the abundance of circulating plasma bile acids. Cosine cycles were observed in Z scored mean time courses of tauro-, glyco-, and unconjugated forms of the primary bile acids, cholic acid (CA) and chenodeoxycholic acid (CDCA), and the secondary bile acids, deoxycholic acid (DCA) and lithocholic acid (LCA), following a cosine pattern across the 36-hour period during the sleep/wake phase. Values are the mean Z scores (shaded area indicates standard deviation) from all DLMO-corrected individual time courses, calculated for each bile acid (DLMO, 21:46 ± 0:29 h:min, mean ± SEM). **C** Cross-correlation analysis indicating phase differences between all bile acid species. Values and red background intensity indicate the magnitude of phase difference in decimal hours. Substantial phase differences can be observed between conjugated and unconjugated bile acids. Source data are provided as a Source Data file.

**Table 1 | Mean acrophase time (decimal h) for plasma bile acids observed to follow cosine dynamics across the 24 h period**

| Bile acid | Class | Conjugation | Entrained protocol | | Entrained with sleep deprivation protocol | | | Constant routine protocol | | | Receptors |
|---|---|---|---|---|---|---|---|---|---|---|---|
| | | | Mean acrophase (h) | SD | Mean acrophase (h) | SD | Phase shift SDvE | Mean acrophase (h) | SD | Phase shift CRvE | |
| CA | Primary | Unconjugated | 6.16 | 0.9 | 5.77 | 0.92 | −0.39 | 9.05 | | | FXR; CAR; TGR5 |
| GCA | Primary | Glycine | 21.81 | 0.47 | 22.19 | 0.42 | 0.38 | 10.17 | | | TGR5; SIP2 |
| TCA | Primary | Taurine | 22.53 | 0.77 | 21.96 | 0.48 | −0.58 | 11.34 | | | TGR5; SIP2 |
| CDCA | Primary | Unconjugated | 7.04 | 0.63 | 5.44 | 1.05 | −1.60 | | | | FXR*; TGR5 |
| GCDCA | Primary | Glycine | 22.36 | 0.53 | 22.4 | 0.34 | 0.04 | | | | FXR; TGR5 |
| TCDCA | Primary | Taurine | 22.32 | 0.71 | 22.18 | 0.44 | −0.13 | | | | FXR; TGR5 |
| DCA | Secondary | Unconjugated | 7.71 | 0.94 | 6.16 | 0.94 | −1.55 | | | | FXR; TGR5 |
| GDCA | Secondary | Glycine | 22.10 | 0.42 | 22.41 | 0.37 | 0.31 | | | | TGR5; SIP2 |
| TDCA | Secondary | Taurine | 22.58 | 0.6 | 22.29 | 0.39 | −0.29 | | | | TGR5; SIP2 |
| LCA | Secondary | Unconjugated | 17.29 | 1.13 | | | | 13.93 | 1.01 | −3.36 | TGR5; PXR; FXR; VDR |
| GLCA | Secondary | Glycine | 18.07 | 1.20 | 19.08 | 1.37 | 1.01 | 16.61 | 0.78 | −1.46 | TGR5 |
| TLCA | Secondary | Taurine | 20.83 | 0.55 | 21.53 | 0.62 | 0.69 | 21.75 | 1.05 | 0.92 | TGR5 |
| UDCA | Secondary | Unconjugated | 8.36 | 0.94 | 6.67 | 0.95 | −1.69 | | | | FXR; TGR5; GR; MR |
| GUDCA | Secondary | Glycine | 21.74 | 0.38 | 22.41 | 0.35 | 0.67 | | | | TGR5 |
| HCA | Secondary | Unconjugated | 7.08 | 0.58 | 6.57 | 0.68 | −0.50 | | | | TGR5 |
| GHCA | Secondary | Glycine | 22.21 | 0.48 | 22.38 | 0.48 | 0.17 | | | | TGR5 |

Data are shown for individuals under the entrained sleep/wake (E), entrained sleep deprivation (SD), and constant routine (CR) protocols. The known receptors for these bile acids are provided. *SD* standard deviation. Mean acrophase calculated from linear mixed-effect cosinor models using individual variability as a random factor. *CA* cholic acid, *GCA* glycocholic acid, *TCA* taurocholic acid; *CDCA* chenodeoxycholic acid, *GCDCA* glycochenodeoxycholic acid, *TCDCA* taurochenodeoxycholic acid, *LCA* lithocholic acid, *GLCA* glycolithocholic acid, *TLCA* taurolithocholic acid, *DCA* deoxycholic acid, *GDCA* glycodeoxycholic acid, *TDCA* taurodeoxycholic acid, *UDCA* ursodeoxycholic acid, *HCA* hyocholic acid, *GHCA* glycohyocholic acid, *FXR* farnesoid nuclear receptor, *CAR* constitutive androstane receptor, *TGR5* takeda G-protein-coupled receptor 5, *SIP2* phingosine-1-phosphate receptor 2, *PXR* pregnane nuclear receptor, *VDR* vitamin D receptor.

$3 \times 30\%$ (meals) and $1 \times 10\%$ (snack). The macronutrient composition of the daily food intake was fat 35%, carbohydrates 49%, protein 12% of energy (breakdown shown in Supplementary Data 2). Plasma melatonin was measured in hourly samples using radioimmunoassay analysis, and dim light melatonin onset (DLMO) time was calculated for each participant using the 25% threshold method as previously described[19]. Mean DLMO was 21:46 h:min (±0:29 h:min, SEM). The bile acid profiles of plasma sampled every 2 hours over a 48-hour period were measured using an ultra-performance liquid chromatography-mass spectrometry (UPLC-MS/MS) based approach[20]. This measured the concentration of the 16 bile acids including the host-derived primary bile acids; cholic acid (CA), glycocholic acid (GCA), taurocholic acid (TCA), chenodeoxycholic acid (CDCA), glycochenodeoxycholic acid (GCDCA) taurochenodeoxycholic acid (TCDCA), hyocholic acid (HCA), glycohyocholic acid (GHCA); and the secondary bile acids that arise from gut microbial metabolism including, deoxycholic acid (DCA), glycodeoxycholic acid (GDCA), taurodeoxycholic acid (TDCA), lithocholic acid (LCA), glycolithocholic acid (GLCA), taurolithocholic acid (TLCA), ursodeoxycholic acid (UDCA), and glycoursodeoxycholic acid (GUDCA). The timing of the samples was individually corrected according to each participant's DMLO, as a reliable marker of circadian phase.

To assess 24 h rhythmicity in the bile acid profiles, we used linear mixed-effect cosinor models. For each individual, the plasma concentration of a bile acid at each specific timepoint was Z scored within the individual across the 48-hour period covering both the entrained and sleep deprivation periods. This was performed separately for each bile acid in each participant. Cosinor fits were statistically compared against the null hypothesis of the constant model across the sleep/wake period, where individual variability was included as a random factor (Fig. 1A, days 2−3). The same approach was used for the sleep deprivation period (Fig. 1A, days 3−4). For a bile acid to be considered rhythmic, a significant cosine fit was required. From this analysis, a significant 24 h cosine rhythm was observed in all 16 bile acids (Fig. 1B;

individual fits shown in Supplementary Fig. S1). The relative proportions of the bile acids within the overall pool across the study are shown for each individual in Supplementary Fig. S2. For the cyclic bile acids, the mean peak phase times of each bile acid are provided in Table 1. The concentration ranges for each bile acid are provided in Supplementary Data 3. These are calculated as the mean minimum and mean maximum concentration of each bile acid across the 24 h period from all participants. TCDCA and GCDCA had the highest maximum circulating concentrations with 9.06 and 2.38 μM, respectively, followed by the unconjugated CA (1.35 μM) and CDCA (1.31 μM).

The rhythmic bile acids included primary and secondary bile acids as well as their conjugated and unconjugated forms. The circulating conjugated bile acids (GCA, GCDCA, TCDCA, GDCA, GLCA, GUDCA, GHCA), derived from the liver, peaked early in the evening (acrophase range: −1.42 to −5.93 h from DLMO; equivalent to -16:00−20:00 h:min) and the unconjugated bile acids, products of bacterial deconjugation, peaked several hours later in the early morning (acrophase range: 6.16−8.36 h from DLMO; -04:00−06:00 h:min). This included both primary (CA, CDCA) and secondary (DCA, UDCA, HCA) unconjugated bile acids. Unconjugated LCA peaked in the plasma at a comparable time to its conjugated forms (17.29 h from DLMO; -15:00). Cholecystokinin is a gut hormone released after a meal that stimulates gall bladder contraction to secrete bile into the intestine. Given the single daily peak in bile acids, with conjugated bile acids increasing in the circulation prior to lunch and peaking after dinner, and the balanced macronutrient and caloric density across the three daily meals, these patterns are unlikely to be driven exclusively by food patterns.

Cross-correlation analysis identified strong similarities between the time courses of the bile acid profiles (mean ± SEM, $R^2 = 0.82 \pm 0.01$ between $16 \times 16$ comparisons), when time-aligned on maximal similarity (Fig. 1C). The strongest correlations were observed within the conjugated bile acids and within the unconjugated bile acids ($R^2 = 0.90 \pm 0.01$ and $0.89 \pm 0.02$, respectively). Lower correlations were observed between conjugated and unconjugated bile acids

($R^2 = 0.75 \pm 0.01$). Consistent with this observation, minimal phase differences in time course profiles were seen within the conjugated bile acids or within the unconjugated bile acids ($51 \pm 10$ mins and $31 \pm 9$ mins, respectively). However, the average circular phase difference between conjugated and unconjugated bile acids was substantial ($8:32 \pm 00:24$ h:min). This likely reflects the different sites of absorption, with most gallbladder-secreted bile acids (i.e. conjugated bile acids) being absorbed in the small intestine, while the unconjugated forms are produced from bacterial deconjugation and absorbed in the colon.

### Bile acid rhythms are associated with rhythms in the plasma lipids

Bile acids have an important role in lipid digestion and absorption and can also serve as signaling molecules to alter biochemical pathways and functions by binding to different nuclear and membrane-bound receptors (Table 1). To establish the potential for these bile acid cycles to impact host lipid metabolism, various lipids from a range of classes (acylcarnitines, lysophosphatidylcholines, phosphatidylcholines, sphingolipids) were measured in these plasma samples using a targeted UPLC-MS/MS-based approach. A range of amino acids and biogenic amines were also measured with this approach. We have previously shown clear diurnal cycles in these metabolites[16]. Cross-correlation analysis was performed to identify time-delayed relationships between the rhythmic bile acids ($n = 16$) and the metabolites ($n = 170$). We observed that the distribution of all significant cross-correlation ($R^2$) values ($N = 2041$ out of 2720) was right-skewed. To account for multiple testing, only those relationships with a significant $R^2$ value above the 95% confidence interval of the mean ($R^2$ cutoff = 0.832) were accepted. In total, 72 temporal relationships between bile acids and metabolites met this criterion (Supplementary Data 4). Bile acids, a mix of unconjugated, taurine, and glycine-conjugated species, were found to be correlated with 20 unique phosphatidylcholines, 7 sphingomyelins, 7 acylcarnitines, 2 lysophosphatidylcholines, and tryptophan, taurine, sarcosine, and dimethylarginine. LCA was correlated with the most features, which included 15 phosphatidylcholines, and 1 acylcarnitine. These lipid species peaked in the blood at the same time as LCA. These associations, along with their phase shifts, are presented in Fig. 2.

### Daily bile acid cycles are maintained with sleep deprivation, but their relationships with circulating lipids are disrupted

Sleep is key for maintaining metabolic homeostasis, and sleep deprivation has been implicated in altered glucose and lipid metabolism[21], reduced energy expenditure, and increased BMI[22]. As these processes can be regulated by the circulating bile acid pool, we examined the effect of sleep on the plasma bile acid profiles. To study this, participants following 24 h of controlled (entrained sleep/wake) laboratory conditions underwent a subsequent 24 h of wakefulness (Fig. 1A). The timing and food content at each set meal was constant between the study days as were the other study parameters (light/dark; posture). No significant change in DLMO was observed between the sleep/wake phase (DLMO, $21:46 \pm 0:29$ h:min, mean $\pm$ SEM) and 24 h of wakefulness (DLMO, $20:21 \pm 1:15$ h:min). Rhythmicity was observed to persist during 24 h wakefulness in all bile acid species during the wake/sleep phase except for LCA (Table 1). Interestingly, plasma taurine concentrations significantly increased during sleep deprivation in these individuals[16]. Negligible phase shifts were noted in the mean acrophase times of the bile acids between the sleep/wake and 24 h wakefulness phases (0.04–1.69 hours; Table 1, Fig. 3A). The peak concentrations of the bile acids during 24 h of wakefulness were comparable with those observed with wake/sleep (Supplementary Data 5). Cross-correlation analysis was performed to assess how the phase relationships between bile acid pairs compared between the entrained sleep/wake conditions versus 24 h wakefulness. This indicated that the phase relationships between all bile acid pairs were maintained with sleep deprivation

except for two pairs whose relationship changed by approximately 3 hours (Fig. 3B). This included LCA and its relationship with UDCA and DCA, where the phase difference between their peaks changed from 6.6 h to 10.2 h and 7.6 h to 10.4 h, respectively. LCA was the only bile acid that lost rhythmicity with sleep deprivation and most likely drove these changes.

Cross-correlation analysis on the plasma bile acid and lipid data from the sleep deprivation segment of the protocol was performed. Changes in the phase relationship between bile acids and lipid time courses are shown in Fig. 3C, D. Most of these relationships did not change in response to sleep deprivation, with 728 relationships (~27%) exhibiting an advance or delay of >6 hours. The histogram of changes in phase relationships shows a second peak, indicating several bile acid-metabolite relationships were shifted with sleep deprivation by approximately 8 hours (Supplemental Table S6). Attributing the changes to specific lipids (thus those species that exhibit a large phase shift in relation to bile acids), exposes a top-quartile contribution of 42 molecules. The four most attributable bile acids were the unconjugated bile acids CA CDCA, DCA, and UDCA.

### The rhythmicity of the bile acids is lost when external cues are kept constant

To assess the influence of exogenous factors such as food, light, and posture on these bile acid dynamics, another human study was performed with 15 males following a 40 h-constant routine protocol where these exogenous factors were removed or kept constant (study protocol shown in Supplementary Fig. S4; Participant characteristics provided in Supplementary Data 1). Here, the participants were subjected to strictly controlled constant routine conditions, including dim lighting (<5 lux in the direction of gaze), semirecumbent posture, and hourly intake of isocaloric snacks with 100 ml of water. This ensured there was no variation in food type or composition across the 40 h constant routine. As with the entrained protocol, these hourly snacks provided 2500 calories across the 24 h period and comprised 50% carbohydrates, 35% fat, and 15% protein. This gold-standard constant routine protocol enabled the influence of the endogenous circadian timing system to be exposed and allowed us to determine whether cyclic patterns in bile acids persist in the absence of external cues. Plasma was sampled hourly for 32 h. DLMO time was calculated for all 15 participants using hourly blood samples (DLMO, $21:50 \pm 0:16$ h:min, mean $\pm$ SEM), and sample collection times were DLMO-corrected for each participant. Bile acids were measured in 2-hourly plasma samples, the abundance of each bile acid was Z-scored within each individual across the sampling period, and linear mixed-effects cosinor models were used to assess rhythmicity. Under constant routine conditions, temporal variation was lost in 10 of the 16 bile acids with only the unconjugated and conjugated forms of cholic acid (CA, GCA, TCA) and lithocholic acid (LCA, GLCA, TLCA) remaining rhythmic (Table 1, Fig. 4). The acrophase of GCA (11.64 h) and TCA (11.19 h) occurred several hours earlier than that observed in the entrained protocol.

## Discussion

In this study, using a combination of highly controlled entrained and constant routine protocols, we demonstrate for the first time in humans that bile acids, host-microbial co-metabolites with the potential to influence the genome and metagenome, follow diurnal rhythms that are primarily driven by the environment. The rhythms in these pan-kingdom metabolites were significantly associated with daily patterns in plasma lipids, including acylcarnitines, phosphatidylcholines, lysophosphatidylcholines, and sphingomyelins, emphasizing the influence of bile acids on the digestion and processing of lipids. The constant routine protocol provides a gold-standard mechanism in humans to assess endogenous circadian rhythms in variables (i.e. those that persist when external cues are removed or kept constant). Reduced rhythmicity of bile acids in the plasma during

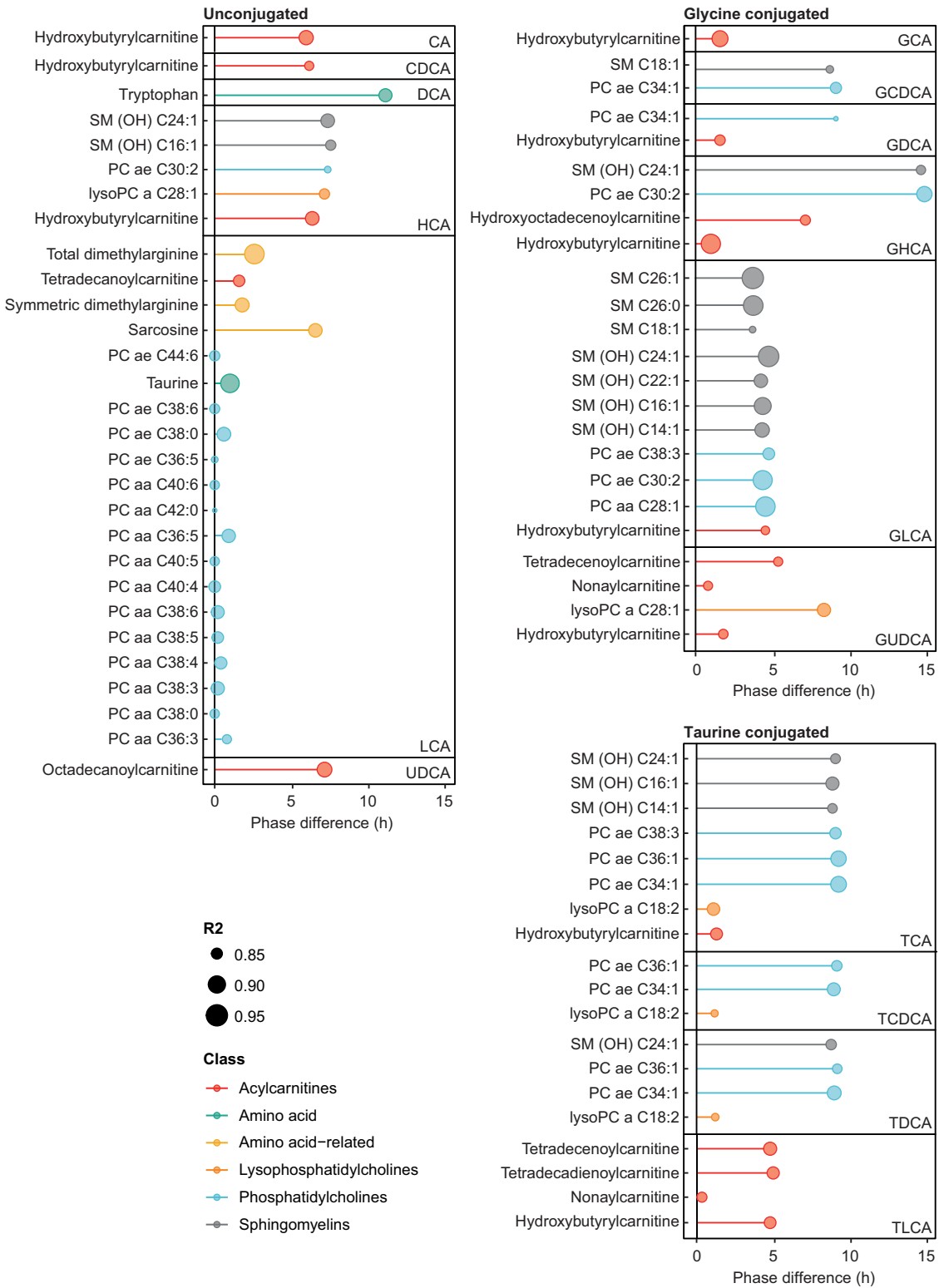

**Fig. 2 | Cross-correlation analysis of circulating rhythmic bile acid species (*n* = 16) with plasma metabolites and lipids in participants following the entrained protocol.** Colors indicate metabolite class, and symbol size indicates correlation strength. The distance of the edge indicates the phase difference between peak bile acid concentration and peak metabolite concentration.

the constant routine protocol confirmed that circulating bile acids are, contrary to previous reports, primarily under environmental control.

All bile acids, including primary, secondary, conjugated, and unconjugated species, exhibited clear daily rhythms in their systemic concentrations. Bile acids are secreted in the gut when meal-associated cholecystokinin stimulates gall bladder contraction. Bile acids then

have an important role in the digestion and absorption of lipids and fat-soluble vitamins. Following reabsorption, bile acids return to the liver via the hepatic portal vein and re-enter the circulation. The observed daily rhythms in these molecules cannot be solely driven by feeding as a single daily acrophase was observed for all bile acids despite each individual consuming three daily meals balanced for

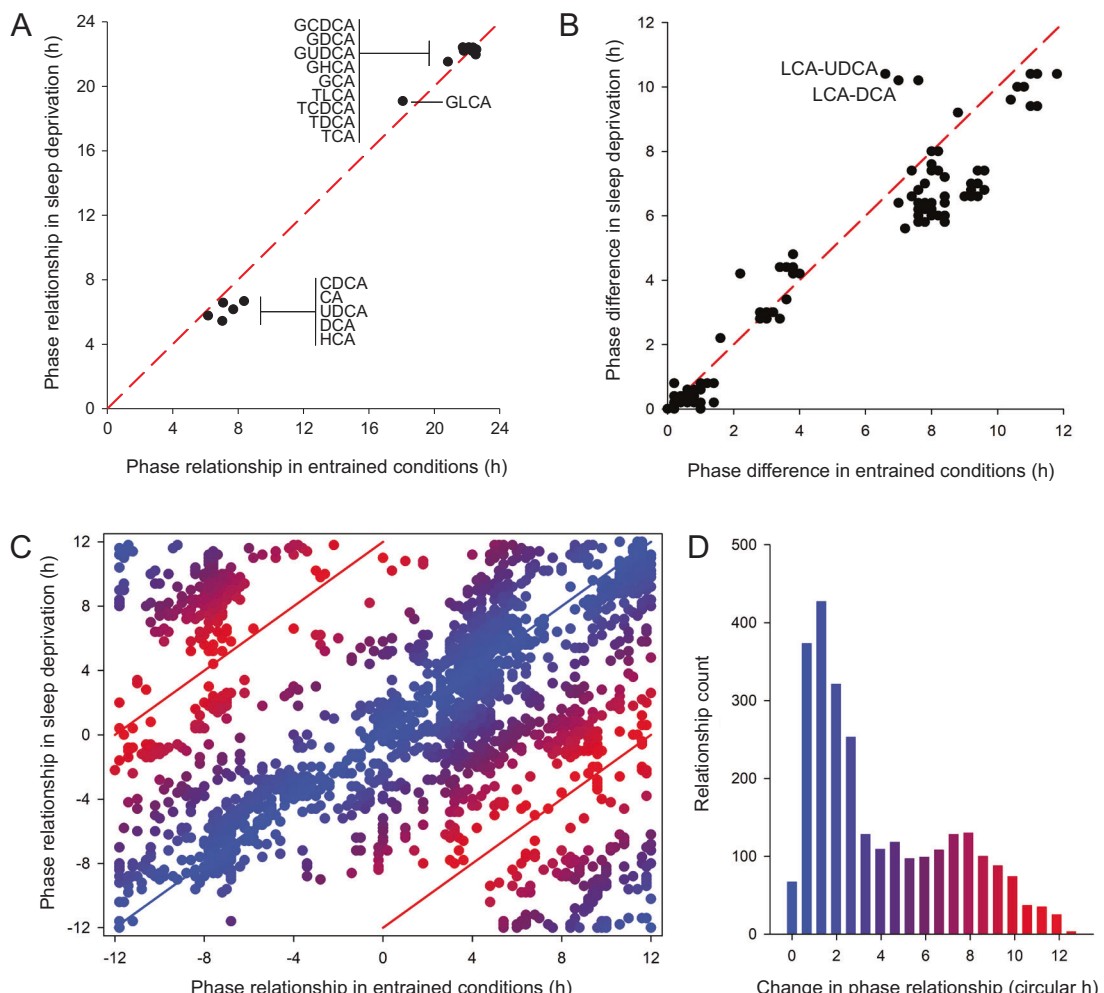

**Fig. 3 | Sleep deprivation does not alter bile acid dynamics, but it disrupts the relationship between plasma bile acids and the circulating metabolome.**
**A** Phase differences in individual bile acids under entrained and sleep deprivation conditions. **B** Phase differences in the cross-correlations between bile acid pairs under entrained and sleep deprivation conditions. All pairs shown including rhythmic and non-rhythmic bile acids. Red dashed line indicates no difference in phase. **C** Phase differences in the relationships between bile acids and metabolites under both conditions. All pairs shown including rhythmic and non-rhythmic bile acids with metabolites and lipids. Blue lines and symbol color indicates no difference, and red lines and symbols indicate a 12-hour shift in the relationship between

conditions. **D** Histogram indicating the frequency of bile acid-metabolite changes by circular hours between the two study conditions. A peak can be observed in 0–2 hours phase shift between the two conditions (i.e., no change) and 7–8 hours phase shift. CA cholic acid, CDCA chenodeoxycholic acid, DCA deoxycholic acid, GCA glycocholic acid, GCDCA glycochenodeoxycholic acid, GDCA glycodeoxycholic acid, GHCA glycohyocholic acid, GUDCA glycoursodeoxycholic acid, HCA hyocholic acid, LCA lithocholic acid, TCA taurocholic acid, TCDCA taurochenodeoxycholic acid, TDCA taurodeoxycholic acid, TLCA taurolithocholic acid, UDCA ursodeoxycholic acid. Source data are provided as a Source Data file.

energy content and macronutrients. Bile acid synthesis has previously been shown to be regulated by several CLOCK-dependent transcription factors. This includes Rev-ERBα (key regulator in cholesterol and bile acid synthesis), DBP (regulates cholesterol 7α-hydroxylase [CYP7A1], the rate-limiting enzyme of the classical bile acid synthesis pathway), and RORα (regulates CYP8B1; involved in CA synthesis)[23–26]. In addition, the bile acids themselves contribute to these patterns through a negative feedback loop, where the transcription of CYP7A1 is inhibited by FXR, which is activated by bile acids. Bile salt export pump (BSEP) and multidrug-resistant-associated protein 2 (MRP2) are responsible for the active transport of bile acids from the canalicular membrane of the hepatocytes to bile. Diurnal expression of MRP2, but not BSEP, the principal bile acid efflux pump, has been observed in mice, peaking in the dark phase when mice are active[27]. Diurnal expression was also noted for the intestinal bile acid transporters, apical sodium-dependent bile salt transporter, and the organic solute transporter α in mice, as well as the sodium-dependent taurocholate

cotransport peptide, which uptakes bile acids into hepatocytes[28]. Thus, while data in mice suggest that the synthesis and transport of bile acids may be regulated by the host circadian system, the observations from this study indicate that the circulating levels, which are partially influenced by gut microbial metabolism, are not. Under entrained conditions, this results in consistent cycles in the release, absorption, and circulation of bile acids throughout the day, but these dynamics are lost when environmental cues are removed.

No differences were seen in the acrophase (peak) timings of primary and secondary (products of bile acid dehydroxylation) bile acid rhythms. However, a substantial phase difference was observed between the conjugated and unconjugated forms. Bile acid coenzyme A:amino acid N-acyltransferase is the enzyme responsible for conjugating bile acids and is regulated by FXR. In mice, the hepatic gene expression of *Fxr* was not observed to exhibit diurnal variation, however, data is lacking for humans[28]. Bile salt hydrolase (BSH) enzymes expressed by bacteria in the gut hydrolyze the amide bond between

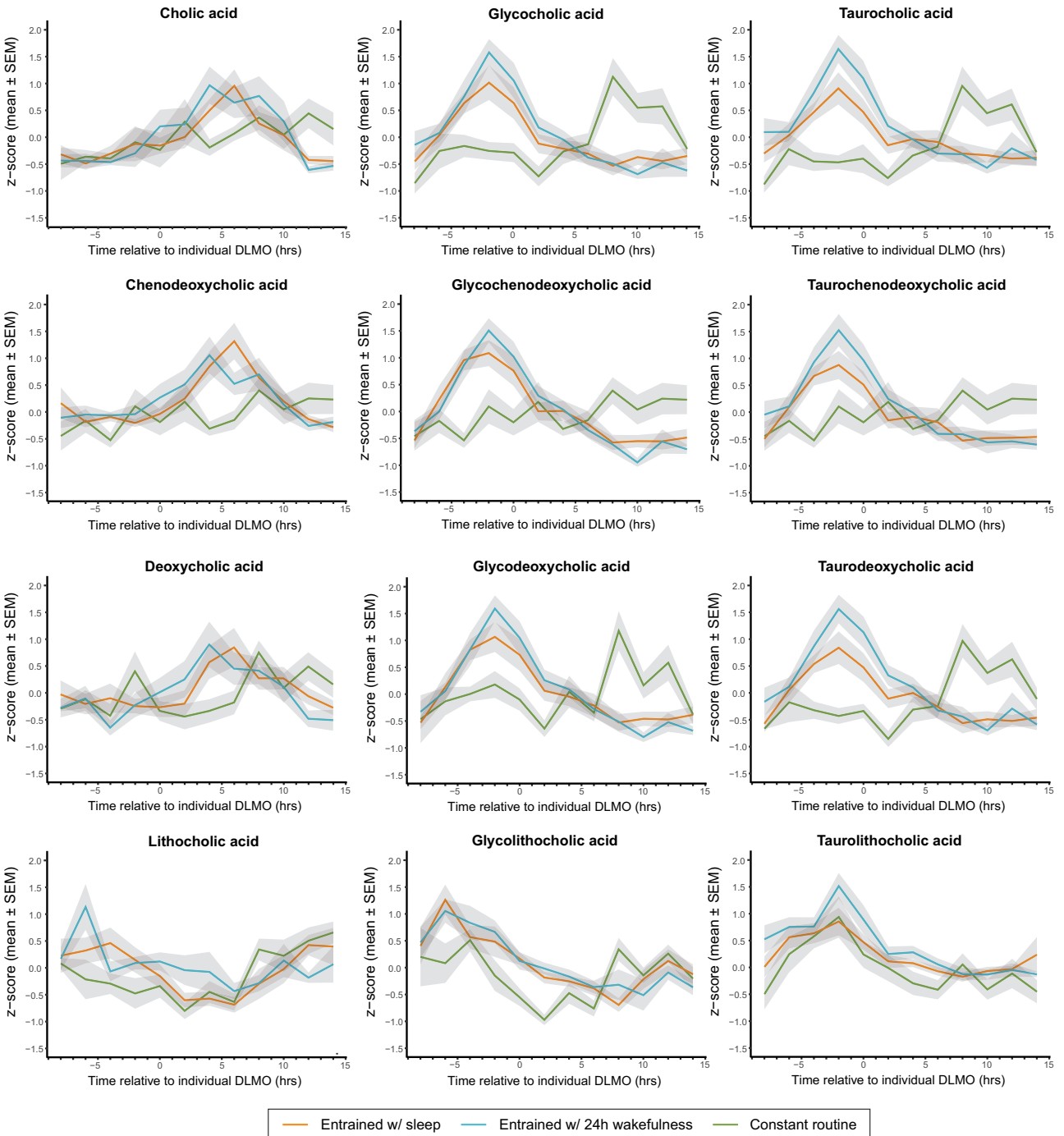

**Fig. 4 | Circulating bile acid rhythmicity is lost in the constant routine protocol that removes the influence of exogenous factors.** Primary and secondary bile acids in their conjugated (glycine- and taurine-) and unconjugated forms are displayed for all individuals from the entrained wake/sleep, entrained with 24-hour wakefulness (sleep deprivation), and constant routine protocols. Values are the mean $Z$ scores (shaded area indicates standard error of the mean) from all DLMO-corrected individual time courses, calculated for each bile acid. Source data are provided as a Source Data file.

the bile acid and its conjugated amino acid, releasing the unconjugated form. Under entrained conditions, the conjugated bile acids peaked in the circulation early in the evening, close to DLMO, whereas the unconjugated bile acids (products of BSH) peaked several hours after this, in the early to late morning. After release into the gut, ~95–99% of conjugated bile acids are reabsorbed in the small intestine, while 1–5% reach the colon and are deconjugated by BSH activity. The phase differences (~6–8 hours) between the conjugated and unconjugated forms likely reflect variation in the sites of absorption and the timing of bacterial BSH activity. The conjugated bile acids in the plasma are

hypothesized to reflect those absorbed in the small intestine, and the unconjugated forms are those modified by bacterial BSH enzymes and absorbed in the colon. Given the single daily peak of these BSH products, this suggests that bacterial BSH activity is most active at night and in the early morning. Based on the current data, we cannot exclude the potential for diurnal variation in bile acid absorption and transport to explain these findings however, diurnal fluctuations in BSH activity have been previously noted in mice[29]. Interestingly, the activity of BSH has been demonstrated to modify the expression of key regulators of lipid metabolism in both the ileum and liver and has been shown to

significantly reduce weight gain, serum low-density lipoprotein cholesterol, and liver triglycerides in mice fed normal and high-fat diets[30]. Conversely, reductions in BSH activity following antibiotic treatment in animals have been observed to promote weight gain[31]. The implications of misalignment between feeding times and BSH activity on lipid processing, whole-body metabolism, and weight gain warrant further investigation.

Daily variation in the bile acid pool may modify its physicochemical and signaling properties with implications for the host. In vitro studies have shown that conjugated bile acids are stronger activators of FXR and TGR5 than unconjugated bile acids[32]. FXR signaling activates genes involved in the mobilization of lipid stores, triglyceride clearance, and the β-oxidation of free fatty acids[33]. This includes the induction of peroxisome proliferator-activated receptor α, which has a critical role in mediating triglyceride metabolism[34]. In the entrained protocol, conjugated bile acids were significantly cross-correlated with several lipid species with modest phase differences. This included acylcarnitines, whose main function is to transport long-chain fatty acids into mitochondria for β-oxidation to provide energy, as well as, phosphatidylcholines, lysophosphatidylcholines, and sphingomyelins. These conjugated bile acids begin to rise in circulation before lunch and peak after dinner. This supports the hypothesis that bile acids, through their signaling roles, prime the metabolic system of the host prior to dietary intake to efficiently process nutritional components, including triglycerides and fatty acids. Furthermore, bile acids have been shown to promote energy expenditure by activating type 2 iodothyronine deiodinase (D2) in brown adipose tissue via TGR5. This was demonstrated by Broeders et al. in humans with CDCA and has been proposed as a potential treatment for obesity and related metabolic diseases[35]. It remains unclear how the potency of this activation varies across different bile acid species or how this activation changes throughout the day, but these findings further support the role of these molecules in tuning the metabolic readiness of the holobiont for receiving and processing energy intake.

Sleep deprivation is closely associated with circadian disruption and has been suggested to induce metabolic dysregulation, which can lead to disorders such as metabolic syndrome and cardiovascular diseases[36]. As a key factor in lipid digestion and metabolic regulation, we explored the impact of 24 h of continuous wakefulness on the circulating bile acid pool. Sleep disruption has previously been shown to suppress *Cyp7a1*, a key bile acid synthesis gene[37]. However, in the current study, sleep deprivation was not observed to disrupt the circulating bile acid rhythms, except for LCA. This indicates that 8 h of acute sleep deprivation is insufficient to induce substantial effects on the bile acid pool and further studies are required to assess the consequences of prolonged insufficient sleep. Despite acute sleep deprivation not altering these bile acid rhythms, it did impact bile acid-lipid relationships. Specifically, associations between bile acids acylcarnitines, and phosphatidylcholine were lost. Given the importance of bile acids in readying the metabolic system for triglyceride clearance and fatty acid oxidation, this dysregulation could have broad health implications for individuals deprived of sleep or shift workers. Interestingly, shift work is associated with dyslipidemia, characterized by increased circulating triglycerides and total cholesterol, particularly in permanent night shift workers[38]. Moreover, a short sleep duration (≤6 h) has been significantly associated with the incidence of obesity[39]. Understanding the implications of these findings is important, particularly given the rising prevalence of shift work globally, and its increased risk for metabolic syndrome, cardiovascular disease, gastrointestinal disorders, and poor mental health, features previously linked with bile acids[18,40–43]. It should be noted that the timing of meals and their macronutrient composition was constant between the entrained period of the study and 24 h wakefulness. To increase our understanding of how bile acids contribute to the adverse health outcomes associated with shift work further studies are required to explore the consequences of varying meal timings and the degree of desynchronization with bile acid cycles. In addition, feeding studies are required to assess such outcomes in shift workers receiving nutritional components that do not require bile acids for their digestion (e.g., vegan diets[44]).

A limitation of the current study was an inability to sample the GI tract at the same temporal resolution as the blood. This prevented the daily variation of the gut microbiota and BSH from being studied. Given the known antimicrobial properties of the bile acid pool it would be interesting to investigate if the observed bile acid dynamics mapped to daily changes in microbial composition and whether these were lost under constant routine conditions. The development of novel sampling methods, such as smart capsules that can sample throughout the gut, may facilitate this in the future.

This study characterizes the daily rhythms of bile acids and pan-kingdom metabolites that serve as a major line of communication linking the intestinal microbiota with the human host genome. These trans-genomic molecules provide a mechanism to prime the host biochemical system to process anticipated nutrients and energy intake and are strongly influenced by environmental cues.

## Methods

The studies were approved by the University of Surrey Ethics Committee and were conducted at the Surrey Clinical Research Centre according to the Declaration of Helsinki and with regard to good clinical practice.

### Materials

Organic solvents (HPLC grade) were used for the sulfation and precipitation, and sodium sulfate was obtained from Sigma Aldrich (Dorset, UK). All mobile phases were prepared with LC-MS grade solvents, formic acid, and ammonium formate from Sigma Aldrich (Dorset, UK). BA standards and deuterated internal standards were obtained from Steraloids (Newport, RI) and Medical Isotopes (Pelham, USA). All the standards were stored at −80 °C.

### Clinical studies

Written and oral consent was obtained from the participants prior to any procedures being performed. Participants were allowed to withdraw from the study at any time. For both studies, participants had to meet defined inclusion/exclusion criteria to be deemed eligible for the study (see Extended Methodology in Supplementary Information). All participant information was coded and held in strictest confidence according to the Data Protection Act (United Kingdom, 1998). For 7 days prior to the in-laboratory session for both studies, individuals maintained regular sleep/wake schedules aligned with their habitual sleep patterns. For the entrained study, participants maintained a 2300–0700 h schedule, and for the constant routine study, individuals selected an 8-h sleep period going to bed between 2200 and 0100 h and waking up between 0600 and 0900 h. Sleep logs, time-stamped voicemail, and activity/light monitors (Actiwatch; CamNtech, Cambridge, UK) were used to confirm compliance. During the final 72 h of this baseline period, participants were requested to refrain from alcohol and caffeine consumption. This baseline period ensured that participants were not sleep-deprived and that their circadian phase was stabilized prior to entering the clinical study.

### Entrained study conditions

The 4-day in-laboratory session (Fig. 1A) was conducted at the Surrey Clinical Research Centre. This included 15 healthy male participants (mean ± SD, age 23.7 ± 5.4 years; BMI 24.9 ± 2.6 kg/m²; sex determined based on self-report). This consisted of an adaptation night (day 1) followed by a 48-h sampling period beginning at 12:00 h,

which comprised a 24-h period (day 2–3) incorporating an 8-h sleep opportunity (23:00–07:00 h; night 2 [N2]), followed by a 24 h period (beginning at 12:00 h, day 3–4) during which participants remained continually awake (day 2/N3). Standardized meals were provided at 07:00, 13:00, and 19:00 h with a snack at 22:00 h; water was available *ad libitum*. A total of 2,500 calories per day was divided into 3 × 30% (meals) and 1 × 10% (snack). The macronutrient composition of the daily food intake was fat 35%, carbohydrates 49%, protein 12% of energy. This was designed to reflect the UK dietary guidelines (fat 35%, carbohydrates 50%, protein 15% of energy). Blood samples were collected for 48 h from 12:00 h on day 2 at hourly intervals for melatonin analysis and 2-hourly intervals for bile acid analysis and targeted metabolomics analysis. Blood samples were collected into lithium heparin tubes and separated by centrifugation. Samples were received on dry ice and stored at −80 °C until needed for preparation and analysis. Further details of this study protocol have been reported elsewhere[45,46].

## Constant routine conditions
The constant routine protocol (Supplementary Fig. 2) included 15 healthy male participants. Following the baseline-at-home period, the participants were admitted into the laboratory, where abstinence from alcohol, nicotine, and drugs of abuse was confirmed. The in-laboratory session included an adaptation night with habitual sleep times (N1) followed by continual wakefulness until 2300 h on day 3. Electroencephalography monitoring occurred from 1200 h on day 2 until 2300 h on day 3 to ensure the participants remained awake throughout the protocol. The participants were subjected to strictly controlled constant routine conditions, including dim lighting (<5 lux in the direction of gaze), semirecumbent posture, and hourly intake of isocaloric snacks with 100 ml of water. Participants were unaware of clock time throughout the study period. Hourly blood samples were collected via an intravenous catheter. Bile acids were measured in 2-hourly blood samples collected from 1500 h on day 2 until 2300 h on day 3.

## Melatonin measurement
To determine concentrations of melatonin, radioimmunoassay analysis was performed on hourly plasma samples (Stockgrand Ltd., University of Surrey) as previously described[19]. The data were used to calculate DLMO, using a defined 25% threshold, for each individual for both the wake/sleep and sleep deprivation periods, as described previously[19,45]. The calculated DLMO was used to phase-adjust the bile acid and metabolite data.

## UPLC-MS bile acid profiling
Plasma samples were combined with ice-cold methanol (1:3 v:v), vortexed, and centrifuged at $14,000 \times g$ for 15 min at 4 °C. The supernatant was then withdrawn into a glass vial and stored at −80 °C for LC-MS analysis. Bile acid quantification was performed on the plasma samples from the entrained study. For these samples, deuterated internal standards were also added to the sample prior to the addition of methanol. Chromatographic separation of bile acids was performed using Waters ACQUITY ultra-performance liquid chromatography (UPLC) (Waters Ltd, Elstree, UK) with a BEH C8 column (1.7 μm, 100 mm × 2.1 mm) maintained at 60 °C. Eluent A consisted of ultra-pure $H_2O$:acetonitrile 100:0.1 (v:v) with ammonium acetate (1 mM) and acetic acid to adjust the pH to 4.15. Eluent B consisted of acetonitrile:2-propanol 1:1 (v:v) mixture. Initially, a flow rate of 0.6 μL/min was used with 90% A, which was then decreased to 65% A at 9.25 mins until 11.5 min. The flow rate was then increased to 0.65 μL/min until 11.8 min, and a final washing step was applied until 15 min. For the quantification of bile acids in plasma from the entrained protocol, the UPLC system was hyphenated with a Waters Xevo TQ-S mass spectrometer (MS) (Waters, Manchester, UK). For the profiling approach used to assess bile acids in plasma from the constant routine study, the UPLC system was coupled to a Xevo G2-S Q-ToF MS. Both MS were equipped with an electrospray ionization source operating in negative ion mode (ESI-). MS settings were as follows: capillary voltage at 1.5 kV, cone voltage at 60 V, source temperature 150 °C, desolvation temperature at 600 °C, desolvation gas flow at 1000 L/h, cone gas flow at 150 L/h.

## UPLC-MS-based metabolomic profiling
Plasma metabolic profiles were measured on a Waters Acquity UPLC system coupled to a Xevo TQ-S MS using the Biocrates *AbsoluteIDQ* p180 targeted metabolomics kit (Biocrates Life Sciences, Innsbruck, Austria). Plasma samples (10 μl) were prepared according to the manufacturer's instructions, adding several stable isotope-labeled standards to the samples prior to the derivatization and extraction steps. Using either UPLC-MS/MS (liquid chromatography/mass spectrometry) or FIA-MS/MS (flow injection analysis/MS), up to 183 metabolites from five different compound classes (namely acylcarnitines, amino acids, biogenic amines, glycerophospholipids, and sphingolipids) can be quantified. Sample order was randomized and 3 levels of quality controls (QC), run on each 96-well plate. The levels of metabolites present in each QC were compared to the expected values, and the percent coefficient of variation (CV%) was calculated. Data were normalized between batches using the results of quality control level 2 (QC2) repeats across the plate ($n = 4$) and between plates using Biocrates METIDQ software (QC2 correction).

## Data analysis
Cosinor and cross-correlation analysis were performed in MATLAB (version R2023b, MathWorks Inc.). Bile acid and metabolite data were DLMO-corrected for each participant. Circadian rhythmicity was determined using linear mixed-effect cosinor models in Matlab[47,48]. In short, for each bile acid, two cosinor models with a level (horizontal) and sloping mesor were fitted using linear mixed-effect models accounting for individual variability as a random effect. Each model was compared to a reduced linear model using a log-likelihood ratio test, and only significant models in which the amplitude confidence interval did not include 0 (zero) were deemed periodic. In instances where both level and sloping mesor cosinor models were periodic, a further log-likelihood ratio test was used to determine whether the reduced model with a level mesor or the model with a sloping mesor should be used. Significant models returned peak time, amplitude, and mesor with variance measures for the population-level fit, as well as participant-specific fits, based on population fit adjusted for each participant's random effect.

To identify time-delayed relationships across the bile acids and between bile acids and metabolites cross-correlation analysis was used. For bile acid-metabolite relationships, we considered only rhythmic bile acids and their relationship to all metabolites/lipids detected. As many statistical tests were performed between the rhythmic bile acids ($n = 16$) and the metabolites ($n = 170$), to account for multiple testing, only those relationships with a significant $R^2$ value above the 95% confidence interval of the mean ($R^2$ cutoff = 0.832) were accepted.

## Reporting summary
Further information on research design is available in the Nature Portfolio Reporting Summary linked to this article.

# Data availability
All data needed to reproduce the results presented here can be found in the manuscript, figures or supplementary materials (Source Data file). Source data are provided with this paper.

# Code availability
No code is available related to these analyses.

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

## Acknowledgements

The authors thank Professor Anne Skeldon (Department of Mathematics, University of Surrey) for writing the cosinor analysis scripts for MatLab. JRS is supported by the NIHR Southampton Biomedical Research Centre, Biotechnology and Biological Sciences Research Council (BB/W00139X/1, BB/N005953/1), and Medical Research Council (MR/W003597/1). We also thank Joo Ern Ang and Sarah K. Davies for data processing; and Daniel Barrett, Cheryl M. Isherwood, and the Surrey Clinical Research Centre medical, clinical, and research teams for their help with the clinical study (supported by the Biotechnology and Biological Sciences Research Council grant (BB/I019405/1) to D.J.S., V.L.R., and F.I.R.).

## Author contributions

D.J.S., V.L.R. designed the laboratory experiments. A.T.B., M.H.S., V.L.R., J.R.S., B.M., F.I.R., N.R.C. acquired the data. A.T.B., M.H.S., N.R.C., D.R.V.D.V., E.A.W., J.R.S. analyzed the data. A.T.B., M.H.S., D.R.V.D.V., E.A.W., J.R.S. generated the figures. A.T.B., D.R.V.D.V., D.J.S., and J.R.S. wrote the paper.

## Competing interests

The authors declare no competing interests.
