## [Peer Review File · Nature Communications]

Exposing 24-hour cycles in bile acids of male humansREVIEWER COMMENTS

Reviewer #2 (Remarks to the Author):

This article by Bello and colleagues constitutes the first comprehensive description of the daily rhythm of blood bile acids in human and their regulation by sleep and feeding rhythms. Using well designed experiments in a highly controlled environment, authors were able to show that 11/16 bile acids (and their derivatives) were rhythmic, with some correlation with other blood lipids. This rhythm was minimally affected by sleep deprivation but abolished by constant routine (feeding). The study is very descriptive but provide new data about bile acid rhythm in human. The article is globally well written, but a few points need clarification.

- The main issue is the method used to analyse rhythmicity. Authors analysed rhythmicity with a cosinor analysis performed on individual z-scores. If data were analysed individually, why not performing the rhythmicity analysis on real values? Why not analysing the entire z-scored data using a mixed linear model, removing the arbitrary "40%" parameter? While the analysis is fine, it should be improved. In addition, the way the data are presented, it does not allow an evaluation of the individual variability. In addition, the entire dataset should be provided to evaluate this variability and allow other scientist to access the data. It is also not clear from where highest concentrations are coming from (line 139), as this does not correspond to any value in the Tables.

- Some statements are somehow overstated. For example, the fact that the bile acids rhythm is independent from the host circadian system is likely exaggerated. The fact that the environment plays an important role in this rhythm does not exclude that the host circadian clock is also important. It could rather explain why there is only one peak of bile acids while volunteers having 4 meals.

- In several paragraphs, statements are not supported by references. For examples, lines 194-197 or 357-374. In addition, some statements like the absence of rhythmicity for Fxr are not supported by abundant literature in mouse, but also new data in human, showing the opposite.

- The number of the Supplementary Tables are different in the manuscript and in the associated Excel file.

Reviewer #3 (Remarks to the Author):

This study examined the diurnal rhythmicity of bile acids, which are liver-produced small molecules that are secreted into the gut to aid in nutrient absorption and also serve as signaling molecules. The study examines bile acid rhythmicity specifically in human plasma. The authors observe rhythmicity in many conjugated and unconjugated bile acids, and additionally notice a shift in the acrophase for conjugated-unconjugated pairs. This rhythmicity is mostly retained if participants are deprived of sleep and kept on a constant typical feeding schedule, which suggests that external factors primarily regulate this rhythmicity. Finally, a study is presented where participants follow a constant routine, in which the participants are kept awake and fed a small amount hourly to uncouple the contribution of feeding time. The authors find that rhythmicity for almost all bile acids are lost under these conditions, strongly suggesting that external factors from an imposed regular schedule are dominant contributors to bile acid rhythmicity.

I read this manuscript with great interest. I especially appreciated the new insight provided regarding the potential interplay of the rhythms of the host and gut microbiota. Advancing the understanding of this relationship could greatly improve the lives of about 15-20% of people who have shift work schedules. The writing is clear, concise, and easy to understand.

Prior to this work, there was already an extensive literature supporting the connection between bile acids and circadian rhythms. It is known that disruption of circadian rhythms leads to altered bile acid metabolism, and as the authors point out, diurnal rhythmicity of bile acids has already been observed in humans. However, this study offers greater resolution and looks more extensively at types of bile acids. I find the tightly-controlled entrained and constant routine schedule experiments to be particularly well done. Publication of these experiments will be impactful because they unambiguously examine the links between bile acids and circadian rhythms.

My suggestions are mostly minor.

One feature of the results that could use some more discussion is the observation that rhythmicity is not uniform. Why are some bile acids not rhythmic? What could this mean

biologically? Is there something unique about these species?

Related to the above comment, many of the bile acids that are statistically determined to not be rhythmic have patterns that look rhythmic to the eye in the manuscript figures. For example, TDCA is deemed not rhythmic, but its abundance plot almost perfectly overlaps with GDCA (which is deemed rhythmic). I think some additional nuance in the discussion could guide the readers interpretation. Even though the total number of individuals did not have a good fit for TDCA rhythmicity, it appears that the highest average abundance points for the TDCA/GDCA pair are similar. What is the confidence that TDCA is not rhythmic vs this data set is too small to show statistical significance? - If the authors are confident in the designations, it is interesting that there is an overrepresentation of tauro-conjugates that are not rhythmic. Why is this potentially?

Similar to the above comment, in the sleep deprivation experiment it is stated that rhythmicity of CDCA and GLCA is lost. This might be statistically true, but if you look at overall trends CDCA is still highest at night in both conditions. Similarly, there is strong overlap in conditions despite failing the specific statistical test used. This also applies to the gain in rhythmicity for TCA, TDCA, and TLCA. The overall trends are examined the conditions look qualitatively similar (e.g. highest levels for each conditions occur at the same time). I think some additional context would be beneficial for the reader here.

Regarding the statement "This included phase differences between the non-rhythmic LCA and its relationship with UDCA and DCA, which changed from 6.6 – 7.6 h to 10.2 – 10.4 h, respectively." I'm not sure what this means since LCA isn't rhythmic. Some additional explanation would be helpful.

In Figure 4, it is stated that CDCDA retains rhythmicity under constant routine conditions. By eye, it looks like rhythmicity is lost (like most other BAs). Is the fit actually significant?

POINT-BY-POINT RESPONSE TO REVIEWERS

We thank the reviewers for their complementary reviews. Their main comments/queries related to the methodology used to study rhythms in the bile acids. To address this, we have applied an improved statistical approach that simultaneously considers data from all individuals. This refined method is described below and in the manuscript. This methodological improvement, as well as other suggestions addressed below, have substantially strengthened the manuscript.

Reviewer #2 (Remarks to the Author):

1-2. The main issue is the method used to analyse rhythmicity. Authors analysed rhythmicity with a cosinor analysis performed on individual z-scores. If data were analysed individually, why not performing the rhythmicity analysis on real values? Why not analysing the entire z-scored data using a mixed linear model, removing the arbitrary “40%” parameter? While the analysis is fine, it should be improved.

We thank the reviewer for this suggestion. To address this point, we have run mixed-linear effects models on each bile acid using the measures from all individuals. As a result, we are no longer fitting individual curves and defining rhythmic bile acids as those significantly fitting curves in >40% participants. From this new analysis, all 16 bile acids measured are observed to be rhythmic under the entrained protocol. We have updated the methods and results to reflect these modifications.

Methods: Page 23, line 587: “Mixed effect cosinor models were fitted across all participants combined for each bile acid using mixed linear modelling, accounting for individual variability as a random effect. The model was statistically compared to their reduced linear model, and only rhythmic models ($P < 0.05$) in which the amplitude was significantly different from 0 were included as periodic cosine fits.”

Results: Page 4, line 94: “To assess 24 h rhythmicity in the bile acid profiles, we used mixed linear cosinor models.”

Page 5, line 97: “Cosinor fits were statistically compared against the null hypothesis of the constant model across the sleep/wake period, where individual variability was included as a random factor (**Figure 1A**, days 2-3). The same approach was used for the sleep deprivation period (**Figure 1A**, days 3-4). For a bile acid to be considered rhythmic, a significant cosine fit was required. From this analysis, a significant 24 h cosine rhythm was observed in all 16 bile acids (**Figure 1B**; individual fits shown in **Supplementary Figure S1**).”

Table 1 has been updated with the results from these models. The legend and footnote has been updated:

Table 1. Mean acrophase time (decimal h) for plasma bile acids observed to follow cosine dynamics across the 24 h period. Data shown for individuals under the entrained sleep/wake (E), entrained-

sleep deprivation (SD), and constant routine (CR) protocols. The known receptors for these bile acids are provided. SD, standard deviation.

Bile acid	Class	Conjugation	Entrained protocol		Entrained with Sleep Deprivation protocol			Constant routine protocol			Receptors
			Mean acrophase (h)	SD	Mean acrophase (h)	SD	Phase shift SDvE	Mean acrophase (h)	SD	Phase shift CRvE	
CA	Primary	Unconjugated	6.16	0.90	5.77	0.92	-0.39	9.05	2.37	2.89	FXR; CAR; TGR5
GCA	Primary	Glycine	21.81	0.47	22.19	0.42	0.38	10.17	1.01	-11.64	TGR5; SIP2
TCA	Primary	Taurine	22.53	0.77	21.96	0.48	-0.58	11.34	0.99	-11.19	TGR5; SIP2
CDCA	Primary	Unconjugated	7.04	0.63	5.44	1.05	-1.60				FXR ¹ ; TGR5
GDCA	Primary	Glycine	22.36	0.53	22.40	0.34	0.04				FXR; TGR5
TCDCA	Primary	Taurine	22.32	0.71	22.18	0.44	-0.13				FXR; TGR5
DCA	Secondary	Unconjugated	7.71	0.94	6.16	0.94	-1.55				FXR; TGR5
GDCA	Secondary	Glycine	22.10	0.42	22.41	0.37	0.31				TGR5; SIP2
TDCA	Secondary	Taurine	22.58	0.60	22.29	0.39	-0.29				TGR5; SIP2
LCA	Secondary	Unconjugated	17.29	1.13				13.93	1.01	-3.36	TGR5; FXR; FXR ¹ ; VDR
GLCA	Secondary	Glycine	18.07	1.20	19.08	1.37	1.01	16.61	0.78	-1.46	TGR5
TLCA	Secondary	Taurine	20.83	0.55	21.53	0.62	0.69	21.75	1.05	0.92	TGR5
UDCA	Secondary	Unconjugated	8.36	0.94	6.67	0.95	-1.69				FXR; TGR5; GR; MR
GUDCA	Secondary	Glycine	21.74	0.38	22.41	0.35	0.67				TGR5
HCA	Secondary	Unconjugated	7.08	0.58	6.57	0.68	-0.50				TGR5
HCA	Secondary	Glycine	22.21	0.48	22.38	0.48	0.17				TGR5

Mean acrophase calculated from mixed linear cosinor models using individual variability as a random factor.

As all 16 bile acids were rhythmic, we have reanalyzed the bile acid-metabolite relationships. This has resulted in updated results for Supplementary Table 4 and that shown in Figure 2 (revised figure shown below). The related text in the manuscript has also been updated.

Figure 2: Cross correlation analysis of circulating bile acid species with plasma metabolites and lipids in participants following the entrained protocol. Colors indicate metabolite class and symbol size indicates correlation strength. Distance of the edge indicates phase difference between peak bile acid concentration and peak metabolite concentration.

3. The way the data are presented, it does not allow an evaluation of the individual variability.

Following the updated analysis, the figures below have been added to the Supplementary Information (Supplementary Figure S1). As requested, these figures show individual variability. This is now referenced in the results section. Page 5, line 105: “individual fits shown in Supplementary Figure 1”.

Supplementary Figure S1: Individual fits of circulating bile acids following daily cycles. Mixed linear cosinor models for each individual bile acid showing overall and individual fits.

Chenodeoxycholic acid

Glycochenodeoxycholic acid

Taurochenodeoxycholic acid

Deoxycholic acid

Glycodeoxycholic acid

Taurodeoxycholic acid

Lithocholic acid

Glycolithocholic acid

Taurolithocholic acid

Ursodeoxycholic acid

Glycoursodeoxycholic acid

Hyocholic acid

Glychocholic acid

4. The entire dataset should be provided to evaluate this variability and allow other scientist to access the data.

The data used for this analysis (raw and Z-scored) has been collated into a supplementary data file (Source Data.xlsx). This has been uploaded to this submission and will be available to readers.

5. It is also not clear from where highest concentrations are coming from (line 139), as this does not correspond to any value in the Tables.

This has now been simplified to report the mean of the minimum concentrations for each bile acid across all participants and then repeated for the maximum concentrations. This information is presented in Supplementary Table 3 and clarified in the text. This has also been performed for the 24h wakefulness protocol (Supplementary Table 5)

Page 5, line 109: “The concentration ranges for each bile acid are provided in **Supplementary Table S3**. These are calculated as the mean minimum and mean maximum concentration of each bile acid across the 24 h period from all participants. TCDCA and GCDCA had the highest maximum circulating concentrations with 9.06 and 2.38 μM , respectively, followed by the unconjugated CA (1.35 μM) and CDCA (1.31 μM).”

Supplementary Table S3: Acrophase time of each circulating bile acid and the mean minimum and maximum concentrations of each bile acid in the entrained protocol. Values are mean acrophase (peak) time in decimal hours relative to DLMO. The minimum and maximum concentration of each bile acid was calculated across the 24 h period for each individual and the means of these values across all individuals are presented.

Bile acids		Acrophase time (dec. h)	Mean minimum concentration (nM)	Mean maximum concentration (nM)
Cholic acid	CA	6.16	45.19	1352.02
Glycocholic acid	GCA	21.82	107.35	770.44
Taurocholic acid	TCA	22.53	7.01	57.13
Chenodeoxycholic acid	CDCA	7.04	58.44	1312.52
Glychenodeoxycholic acid	GCDCA	22.36	413.56	2382.66
Taurochenodeoxycholic acid	TCDCA	22.32	1614.52	9061.47
Deoxycholic acid	DCA	7.71	156.26	931.32
Glycodeoxycholic acid	GDCA	22.1	125.31	1015.05
Taurodeoxycholic acid	TDCA	22.58	13.01	119.43
Lithocholic acid	LCA	17.29	3.38	33.23
Glycolithocholic acid	GLCA	18.07	2.77	30.31
Tauroolithocholic acid	TLCA	20.83	0.13	6.36
Ursodeoxycholic acid	UDCA	8.36	31.41	154.82
Glycoursodeoxycholic acid	GUDCA	21.74	53.81	273.56
Hyochoolic acid	HCA	7.08	1.53	24.89
Glychoyochoolic acid	GHCA	22.21	1.60	16.63

Supplementary Table S5: Acrophase time of each circulating bile acid and the mean minimum and maximum concentrations of each bile acid during 24 hours of wakefulness. Values are mean acrophase (peak) time in decimal hours relative to DLMO. The minimum and maximum concentration of each bile acid was calculated across the 24 h period for each individual and the means of these values across all individuals are presented.

Bile acids		Acrophase time (dec. h)	Mean minimum concentration (nM)	Mean maximum concentration (nM)
Cholic acid	CA	5.77	52.92	1407.76
Glycocholic acid	GCA	22.19	129.76	982.52
Taurocholic acid	TCA	21.96	10.41	102.02
Chenodeoxycholic acid	CDCA	5.44	40.21	1146.30
Glycochenodeoxycholic acid	GCDCA	22.4	453.21	2538.31
Taurochenodeoxycholic acid	TCDCA	22.18	2017.74	10939.87
Deoxycholic acid	DCA	6.16	148.20	892.07
Glycodeoxycholic acid	GDCA	22.41	138.65	993.18
Taurodeoxycholic acid	TDCA	22.29	14.45	131.60
Glycolithocholic acid	GLCA	19.08	2.99	33.30
Taurolithocholic acid	TLCA	21.53	0.42	7.02
Ursodeoxycholic acid	UDCA	6.67	25.99	161.16
Glycoursodeoxycholic acid	GUDCA	22.41	62.31	304.06
Hyochoolic acid	HCA	6.57	1.60	25.56
Glychoyochoolic acid	GHCA	22.38	1.58	18.95

6. Some statements are somehow overstated. For example, the fact that the bile acids rhythm is independent from the host circadian system is likely exaggerated. The fact that the environment plays an important role in this rhythm does not exclude that the host circadian clock is also important. It could rather explain why there is only one peak of bile acids while volunteers having 4 meals.

This is a valid comment. We have updated the manuscript throughout to emphasise that the environment is a stronger determinant of the rhythms observed in the circulating bile acid signatures rather than the circadian system not having an influence.

Abstract: “We also highlight that bile acid rhythmicity is predominantly lost when environmental timing cues are held constant. This indicates that the environment is a stronger determinant of these temporal dynamics than the intrinsic circadian system of the host.”

Page 3, line 62: “These findings establish that the environment exerts a stronger influence on the circulating bile acids, key trans-genomic signaling molecules that impact on both the host and microbiota, than the intrinsic circadian timing system of the host.”

Page 14, line 281: “In this study, using a combination of highly controlled entrained and constant routine protocols we demonstrate for the first time in humans that bile acids, host-

microbial co-metabolites with the potential to influence the genome and metagenome, follow diurnal rhythms that are primarily driven by the environment. independent from the host circadian system.”

Page 14, line 290: *“Reduced rhythmicity of bile acids in the plasma during the constant routine protocol confirmed that circulating bile acids are, contrary to previous reports, primarily under environment control but are driven by exposure to environmental factors.”*

7. In several paragraphs, statements are not supported by references. For examples, lines 194-197 or 357-374.

References have now been added where appropriate to support statements.

Page 14, line 309: *“Diurnal expression of MRP2, but not BSEP the principal bile acid efflux pump, has been observed in mice, peaking in the dark phase when mice are active¹.”*

²⁷Zhang, Y.K.J., Guo, G.L., and Klaassen, C.D. (2011). Diurnal Variations of Mouse Plasma and Hepatic Bile Acid Concentrations as well as Expression of Biosynthetic Enzymes and Transporters. PLoS One 6, e16683.

Page 16, line 350: *“FXR signaling activates genes involved in the mobilization of lipid stores, triglyceride clearance, and the β -oxidation of free fatty acids³³.”*

³³Almeqdadi M. and Gordon F.D. (2024). Farnesoid X Receptor Agonists: a Promising Therapeutic Strategy for Gastrointestinal Diseases. Gastro Hep Advances 3(3): 344-352.

Page 16, line 352: *“This includes the induction of peroxisome proliferator-activated receptor α (PPAR α), which has a critical role in mediating triglyceride metabolism³⁴.”*

³⁴Torra I.P., Claudel T., Duval C., Kosykh V., Fruchart J.C. and Staels B. (2003) Bile acids induce the expression of the human peroxisome proliferator-activated receptor alpha gene via activation of the farnesoid X receptor. Mol Endocrinol. 17(2): 259-272.

8. The number of the Supplementary Tables are different in the manuscript and in the associated Excel file.

This has now been corrected.

Reviewer #3 (Remarks to the Author):

My suggestions are mostly minor.

Q1-4. One feature of the results that could use some more discussion is the observation that rhythmicity is not uniform. Why are some bile acids not rhythmic? What could this mean biologically? Is there something unique about these species? Many of the bile acids that are statistically determined to not be rhythmic have patterns that look rhythmic to the eye in the manuscript figures. What is the confidence that TDCA is not rhythmic vs this data set is too small to show statistical significance? If the authors are confident in the designations, it is interesting that there is an overrepresentation of tauro-conjugates that are not rhythmic. Why is this potentially? In the sleep deprivation experiment it is stated that rhythmicity of CDCA and GLCA is lost. This might be statistically true, but if you look at overall trends CDCA is still highest at night in both conditions. Similarly, there is strong overlap in conditions despite failing the specific statistical test used. This also applies to the gain in rhythmicity for TCA, TDCA, and TLCA. The overall trends are examined the conditions look qualitatively similar (e.g. highest levels for each conditions occur at the same time).

These were valid observations/questions from the reviewer. Under the previous cosinor approach considering individuals in isolation and requiring >40% individuals to demonstrate rhythmicity for a bile acid to be considered rhythmic, some bile acids that were visually rhythmic were not considered so statistically. However, following our updated analysis considering all participants simultaneously, all 16 measured bile acids now exhibit rhythmicity under entrained conditions and only lithocholic acid loses rhythmicity following 24 h of wakefulness. As such, these valid questions posed by the reviewer no longer need to be addressed.

5. Regarding the statement “This included phase differences between the non-rhythmic LCA and its relationship with UDCA and DCA, which changed from 6.6 – 7.6 h to 10.2 – 10.4 h, respectively.” I’m not sure what this means since LCA isn’t rhythmic. Some additional explanation would be helpful.

This refers to cross-correlation analysis which explores the similarity of two sets of time-series data (e.g., bile acids) relative to one another. This objectively measures how well the data move in parallel and the time difference (phase shift) in their maximum correlation. Here, the cross-correlation between nearly all bile acid pairs and their phase shifts were comparable between the entrained and sleep deprivation protocols except for LCA-UDCA and LCA-DCA, which were increased by ~3 hours. LCA was the only bile acid noted to lose rhythmicity with sleep deprivation and this impact of 24h wakefulness on LCA most likely explains the change in these phase differences.

This sentence has now been updated to assist the reader:

Page 10, line 212: “*Cross-correlation analysis was performed to assess how the phase relationships between bile acid pairs compared between the entrained sleep/wake*

conditions versus 24h wakefulness. This indicated that the phase relationships between all bile acid pairs were maintained with sleep deprivation except for two pairs whose relationship changed by approximately 3 hours (**Figure 3B**). This included LCA and its relationship with UDCA and DCA, where the phase difference between their peaks changed from 6.6 h to 10.2 h and 7.6 h to 10.4 h, respectively. LCA was the only bile acid that lost rhythmicity with sleep deprivation and most likely drives these changes.”

6. In Figure 4, it is stated that CDCDA retains rhythmicity under constant routine conditions. By eye, it looks like rhythmicity is lost (like most other BAs). Is the fit actually significant?

Using the updated mixed-linear effects model approach, CDCA is not rhythmic under constant routine conditions. This information has been added to Table 1 (see Reviewer 2; Q1) and the text has been updated to reflect this.

Page 12, line 263: “Bile acids were measured in 2-hourly plasma samples, the abundance of each bile acid was Z-scored within each individual across the sampling period, and mixed-linear effects models were used to assess rhythmicity. Under constant routine conditions, temporal variation was lost in 10 of the 16 bile acids with only the unconjugated and conjugated forms of cholic acid (CA, GCA, TCA) and lithocholic acid (LCA, GLCA, TLCA) remaining rhythmic (**Table 1, Figure 4**). The acrophase of GCA (11.64 h) and TCA (11.19 h) occurred several hours earlier than that observed in the entrained protocol.”

REVIEWERS' COMMENTS

Reviewer #2 (Remarks to the Author):

The authors have properly addressed all the issues raised by the reviewers. They have also made the raw data available. The article should now be accepted for publication.

Reviewer #3 (Remarks to the Author):

The authors have significantly revised their manuscript in response to reviewer comments. My main comments were addressed through the updated statistical analysis. I have no new concerns with this manuscript upon review of the revisions.